# Association of Handgrip Strength with Diabetes Mellitus in Korean Adults According to Sex

**DOI:** 10.3390/diagnostics12081874

**Published:** 2022-08-02

**Authors:** Sung-Bum Lee, Ji-Eun Moon, Jong-Koo Kim

**Affiliations:** 1Department of Family Medicine, Soonchunhyang University Bucheon Hospital, Bucheon 22972, Korea; sblee@schmc.ac.kr; 2Department of Medicine, Graduate School, Yonsei University Wonju College of Medicine, Wonju 26426, Korea; 3Department of Biostatistics, Clinical Trial Center, Soonchunhyang University Bucheon Hospital, Bucheon 14584, Korea; moon6188@schmc.ac.kr; 4Department of Family Medicine, Yonsei University Wonju College of Medicine, Wonju 26426, Korea; 5Institute of Global Health Care and Development, Wonju 26426, Korea

**Keywords:** handgrip strength, DM, gender difference

## Abstract

Diabetes mellitus (DM) is known to lead to many diseases such as cardiovascular disease and chronic kidney diseases. Therefore, it is essential to find diagnostic tools to prevent DM. This study aimed to find the association between handgrip strength and the prevalence of diabetes mellitus (DM) in Korean adults with respect to sex and menopause. A total of 26,536 participants (12,247 men, 6977 premenopausal women, and 7312 postmenopausal women) aged >19 years were recruited. The study population was divided into quartiles of relative handgrip strength. Logistic regression was used to analyse the association between relative handgrip strength and the prevalence of DM. Compared with the lowest quartile, the odds ratio (95% confidence interval (CI)) the prevalence of DM for the fourth quartile (Q4) was 0.57 (0.43–0.75) after adjusting for confounding factors in men; 0.33 (0.14–0.75), premenopausal women; and 0.82 (0.63–1.07), postmenopausal women. The prevalence of DM decreased as relative handgrip strength increased. This inverse association was more significant in men and premenopausal women than that in postmenopausal women.

## 1. Introduction

The prevalence of DM has been rapidly increasing worldwide with an increase in 39 million patients between 2019 and 2021. Approximately 463 million patients with DM have been reported in 2021, and this number is expected to increase to 700 million by 2045 [1]. Many studies have reported that DM is an important cause of chronic kidney disease, cardiovascular disease, and other complications that affect the quality of life, which can induce a substantial socioeconomic burden on the healthcare system [2,3]. Accordingly, it is essential to screen promptly, prevent DM, and reduce DM-related healthcare burdens.

Sarcopenia, a condition characterised by declining muscle strength and mass, is a primary health concern worldwide. The worldwide prevalence of sarcopenia ranges from 10 to 40% [4]. Moreover, it can independently increase cardiovascular diseases, metabolic diseases, and mortality [5,6]. Timely detection of sarcopenia is essential to prevent these comorbidities. Handgrip strength is an inexpensive and simple assessment tool to measure muscle strength and is suitable for the diagnosis of sarcopenia [7]. Previous studies found handgrip strength to be a predictor of non-alcoholic fatty liver disease and healthy aging [8,9].

There have been many studies on the association between handgrip strength and DM; however, there are conflicting reports. Some studies have demonstrated an inverse relationship between handgrip strength and DM [10,11], whereas other studies have found an insignificant association between handgrip strength and DM [12,13]. Furthermore, although handgrip strength is generally affected by body size (height, weight, and body mass index (BMI)), few studies have used normalised handgrip strength to determine the association between handgrip strength and DM. Recent studies have shown that BMI-adjusted relative handgrip strength (RGS) is a more useful predictor than absolute handgrip strength for determining the association between handgrip strength and metabolic disease [14,15]. Therefore, we aimed to investigate the relationship between RGS and DM in Korean adults using nationwide data. Moreover, we assessed the relationship with respect to men, premenopausal women, and postmenopausal women to clarify physiological differences by sex and menopause.

## 2. Materials and Methods

### 2.1. Study Population

In the cross-sectional study, data were collected from the Korea National and Health Examination Survey (KNHANES) from 2014 to 2018. The KNHANES is a national surveillance conducted annually to evaluate the health and nutritional status of Koreans by the Korea Centres for Disease Control and Prevention in the Ministry of Health and Welfare, which has been previously reported in detail [16].

In total, 32,704 participants were recruited between 2014 and 2018. Among the following data, we excluded those who met the following criteria: (1) age <19 years, (2) absence of handgrip strength data, and (3) those who did not answer the questionnaire on DM diagnosis and medication, and those who had missing data regarding fasting glucose and glycosylated haemoglobin (HbA1c) (Figure 1).

The use of KNHANES (2014–2018) data was approved by the Institutional Review Board (IRB) of the Korea Centres for Disease Control and Prevention (IRB No. 2013-12EXP-03-5C, 2015-01-02-6C, and 2018-01-03-P-A). This study was conducted in accordance with the Declaration of Helsinki. Informed consent was obtained from all the participants. The original data are publicly available for free on the KNHANES website (http://knhanes.cdc.go.kr, accessed on 27 July 2022) for academic research.

### 2.2. Measurement of Handgrip Strength

Handgrip strength was assessed three times using a digital grip strength dynamometer (TKK 5401; Takei Scientific Instruments Co., Ltd., Tokyo, Japan) [17]. Trained technicians instructed the participants to stand with their arms in full extension. The participants were instructed to squeeze the dynamometer with as much force as possible and to conduct three attempts per hand, with a minute of rest between each attempt. Absolute handgrip strength was presented as the summation of the maximum value from each hand and was expressed in kilograms. RGS was used to reduce the impact of body size. RGS is defined as the absolute handgrip strength divided by BMI because it was previously used as an indicator of muscle strength [18]. RGS was subdivided into sex-specific quartiles. Moreover, we also used the average value of handgrip strength in order to be compared with the maximum value. The prevalence of DM by using logistic regression with the relative handgrip strength (the summation of the average value from each hand/BMI). The results were presented as Appendix A.

### 2.3. Anthropometric and Laboratory Measurements and General Data

All anthropometric, demographic, and lifestyle data and blood samples were collected from all the participants. The anthropometric data included age, sex, waist circumference (WC), body mass index (BMI), systolic blood pressure (SBP), and diastolic blood pressure (DBP). WC was measured using flexible tape (Seca 220; Seca) at the midpoint between the lowest margin of the rib and the uppermost border of the iliac crest during expiration [19]. The BMI was calculated as weight (kg)/height (m^2^). Weight was measured in kg to the nearest 0.01 kg, and height was measured to the nearest 0.1 cm. Blood pressure (BP) was measured using a mercury sphygmomanometer after the participants rested for 5 min in a sitting position (Baumanometer Wall Unit 33(0850)). All BP examinations were performed on the right arm three times using the same tool at 30 s intervals [20]. Hypertension was diagnosed as SBP ≥ 140 mmHg, DBP ≥ 90 mmHg, or the use of antihypertensive medications [21]. The medication history was obtained using questionnaires. We also obtained demographic and lifestyle data based on the questionnaires that were used to obtain information on age, sex (men, premenopausal and postmenopausal women), smoking history, alcohol uptake, and regular exercise. Based on the questionnaire, smokers were categorised as current smokers, ex-smokers, and non-smokers. Current smokers were defined as those who checked the response “I have smoked more than 5 packs of cigarettes in a lifetime and still smoke.” Ex-smokers were defined as those who checked the response “I have smoked more than 5 packs of cigarettes but do not smoke anymore.” Non-smokers were defined as those who checked the response “I have smoked less than 5 packs of cigarettes in a lifetime.” [22]. Alcohol uptake was defined as drinking at least once a week; alcohol uptake per week was >140 g for men and >70 g for women [23]. Regular exercise was defined as engaging in either moderate or vigorous physical activity more than 3 times per week. The global physical activity questionnaire (GPAQ) was used to evaluate the level of physical activity [24]. Fasting blood glucose (FBG), total cholesterol (TC), triglyceride (TG), alanine aminotransferase (ALT), and aspartate aminotransferase (AST) levels were measured using a Hitachi 7600 Automatic Analyser (Hitachi, Tokyo, Japan) [25]. HbA1c was measured using high-performance liquid chromatography with an automated HGLC-723G7 analyser (Tosoh Corporation, Tokyo, Japan) [25]. 

### 2.4. Definition of Diabetes Mellitus

Diabetes was diagnosed if one of the following WHO criteria was met: 8 h fasting blood glucose (FBG) ≥ 126 mg/dL (7.0 mmol/L), HbA1c ≥ 6.5% (48 mmol/mol), or 2 h glucose level ≥ 200 mg/dL (11.1 mmol/L) after a 75 g oral glucose tolerance test (OGTT) [26]. Participants who reported taking diabetes medication or injecting insulin were also considered diabetic. 

### 2.5. Statistical Analysis

All variables were analysed using independent *t*-test and analysis of variance (ANOVA) test for continuous variables and chi-square tests for categorical variables. Continuous and categorical variables are presented as mean ± standard deviation and *n* (%), respectively. Multivariate binary logistic regression analysis was applied to assess the relationship between RGS (per 0.1 kg) and the prevalence of DM after adjusting for age, WC, regular exercise, smoking history, alcohol uptake, fasting blood glucose, TC, TG, and SBP. RGS values were categorised into quartiles: Q1, ≤2.84; Q2, 2.85–3.30; Q3, 3.31–3.75; Q4 > 3.75 in men; Q1, ≤1.90; Q2, 1.91–2.22; Q3, 2.23–2.53; Q4 > 2.53 in premenopausal women, and Q1 ≤ 1.48; Q2, 1.49–1.81; Q3, 1.82–2.13; Q4 > 2.13 in postmenopausal women. The weakest RGS group (Q1) was used as the reference group. Binary logistic regression was conducted to measure the odds ratios (ORs) and 95% confidence intervals (95% CIs) of DM for the RGS quartiles after adjusting for confounding factors. Weight values were applied to all the variables. The weight values were calculated for the participants to represent the Korean population by accounting for the non-response, complex survey design, and poststratification [16]. *p*-values < 0.05 were considered statistically significant. Pseudo-R2 was used as model diagnostics in order to validate logistic regression analysis (Appendix A). Furthermore, receiver operating characteristic (ROC) curve and area under the curve (AUC) were illustrated to present the predictive power for DM according to RGS (per 0.1 kg) in Appendix A. Statistical analyses were performed using IBM SPSS Statistics (version 22.0; IBM Corp., Armonk, NY, USA).

## 3. Results

### 3.1. Baseline Characteristics of the Study Population

Baseline characteristics of the study population according to the RGS quartiles are shown in Table 1. A total of 26,536 participants (12,247 men, 6977 premenopausal women, and 7312 postmenopausal women) were included in our study. The mean values of some variables significantly reduced with increasing RGS quartile. These variables were age, WC, BMI, FBG, AST, ALT, SBP, and hypertension in men; age, WC, BMI, FBG, TC, TG, AST, ALT, SBP, DBP, and hypertension in premenopausal women; and age, WC, BMI, FBG, SBP, and hypertension in postmenopausal women. Other values that increased more in Q2 than in Q1 but decreased gradually from Q2 to Q4 were TC, TG, and DBP in men and TC, AST, and ALT in postmenopausal women. In contrast, the mean value of regular exercise increased in all groups along with the quartiles.

### 3.2. Association between RGS and the Prevalence of DM

The prevalence of DM decreased with increasing RGS quartiles in both the sexes (Figure 2). This suggests a dose–response relationship between RGS and DM. The association between RGS (per 0.1 kg) and the prevalence of DM in Koreans is shown in Table 2. A higher RGS was inversely associated with the prevalence of DM in all men and premenopausal women models. In contrast, RGS was not significantly related to the prevalence of DM in model 3 in postmenopausal women. 

Table 3 presents the odds ratios (ORs) and 95% confidence intervals (CI) for the prevalence of DM according to the RGS quartiles. The lowest quartile (Q1) of RGS was defined as the reference group [27]. Compared with the reference group, the ORs (95% CI) for DM of the participants in Q4 were 0.16 (0.13–0.21) in men, 0.07 (0.03–0.15) in premenopausal women, and 0.29 (0.23–0.36) in postmenopausal women, when unadjusted. After adjusting for age (Model 1), the ORs for DM in Q4 were 0.32 (0.25–0.41) in men, 0.08 (0.04–0.18) in premenopausal women, and 0.44 (0.34–0.56) in postmenopausal women, which gradually reduced; nevertheless, they were statistically significant. After further adjusting Model 1 for WC, regular exercise, smoking history, and alcohol uptake (Model 2), the ORs (95% CI) in Q4 were 0.54 (0.42–0.71) in men, 0.30 (0.14–0.66) in premenopausal women, and 0.81 (0.62–1.05) in postmenopausal women, which were statistically significant in men and premenopausal women, but insignificant in postmenopausal women. After further adjusting Model 2 for TC, TG, AST, ALT, and SBP (Model 3), the ORs (95% CI) in Q4 were 0.57 (0.43–0.75) in men, 0.33 (0.14–0.75) in premenopausal women, and 0.82 (0.63–1.07) in postmenopausal women. The differences in men and premenopausal women showed statistical significance; however, those in postmenopausal women were insignificant, which was the same trend as that of Model 2.

We have also showed the ORs for the prevalence of DM according to the RGS quartiles after stratified by age; less than 40 years old group, 40–59 years old group, and 60 years old or more group (Appendix A). The lowest quartile (Q1) of RGS was defined as the reference group. Compared with the reference group, the ORs (95% CI) for DM of the subjects in Q4 were 0.21 (0.07–0.66) in men <40 years old, 0.49 (0.33–0.71) in men 40–59 years old, and 0.74 (0.49–1.12) in men ≥60 years old after adjusting Model 3. The ORs (95% CI) for DM of the participants in Q4 were 0.54 (0.12–2.34) in premenopausal women <40 years old, and 0.20 (0.07–0.55) in premenopausal women 40–59 years old after adjusting Model 3. The ORs (95% CI) for DM of the participants in Q4 were 0.89 (0.53–1.49) in in postmenopausal women 40–59 years old, 0.84 (0.61–1.15) in postmenopausal women ≥60 years old after adjusting Model 3. The association of handgrip strength with DM is more significant with the younger group in men. The relationship was also significant in premenopausal women who are 40–59 years old. However, those in postmenopausal women were insignificant. 

ROC curve and AUC were conducted to predict DM according to RGS in Appendix A. AUCs were 0.671 (0.658–0.684) in men in the unadjusted model, 0.804 (0.794–0.814; *p*-value < 0.001) in men after adjusting Model 3, 0.711 (0.677–0.745) in premenopausal women in the unadjusted model, 0.887 (0.866–0.908; *p*-value < 0.001) in premenopausal women after adjusting Model 3, 0.637 (0.621–0.654) in postmenopausal women in the unadjusted model, and 0.772 (0.758–0.785; *p*-value < 0.001) in postmenopausal women after adjusting Model 3.

## 4. Discussion

In a nationwide cross-sectional study performed over 5 years, RGS was negatively associated with the prevalence of diabetes mellitus. Handgrip strength was an independent indicator of DM, irrespective of age, WC, regular exercise, smoking history, alcohol uptake, fasting glucose, TC, TG, AST, ALT, and SBP (in men and premenopausal women), and age (in postmenopausal women). Moreover, there was a dose-dependent relationship between handgrip strength and the prevalence of diabetes. 

Handgrip strength is a well-known indicator of metabolic syndrome [27]. Several studies have investigated the association between handgrip strength and diabetes. However, a previous cross-sectional study that enrolled 8208 participants for 2 years suggested that handgrip strength was inversely related to the prevalence of diabetes [11]. Leong et al., in a prospective urban–rural epidemiology (PURE) study, reported no significant association between grip strength and incident diabetes after adjusting for DM-related factors (BMI, waist-to-hip ratio) [12]. However, we found an association between handgrip strength and diabetes in large number of participants (>30,000) over 5 years. Moreover, the association was still found to be significant despite adjusting for WC, fasting glucose, TG, and blood pressure, which are essential components of metabolic syndrome. Especially, we found a significant association not only in men but also in premenopausal women by classifying women on the basis of menopause status.

Handgrip strength is a useful tool to assess muscle strength because it is inexpensive, quick, and easy to measure [28]. Moreover, low muscle strength is considered to be a principal determinant of sarcopenia rather than low muscle mass because muscle strength is a more important indicator predicting falls, fracture, and all-cause mortality rather than muscle mass [12,29]. Consequently, handgrip strength is generally used as a diagnostic approach for sarcopenia. 

Even though a mechanistic link between handgrip strength and diabetes has not been fully elucidated, we can identify putative mechanisms through mediators related to handgrip strength and diabetes. Low muscle strength is associated with inflammation, which is an important factor in insulin resistance [30]. Decreased muscle strength is related to increased levels of inflammatory markers (tumour necrosis factor-alpha (TNF-α), interleukin-6 (IL-6), and C-reactive protein (CRP)), which can induce the development of diabetes [31,32]. 

Inflammation has also been associated with menstruation. In premenopausal women, TNF-α, IL-6, and CRP levels fluctuate throughout the menstrual cycle. Inflammatory biomarkers increase during ovulation and peak during menstruation [33]. Furthermore, inflammatory factors can contribute to ovarian aging and to the timing of menopause. TNF-α plays an essential role in follicle recruitment and atresia, indicating that this marker can affect menopause [34]. TNF-α receptor levels are associated with the risk of early menopause [35]. In other words, menopause is associated with diabetes through the mediators. Therefore, the prevalence of DM in postmenopausal women was higher than that in premenopausal women, compared with each RGS quartile. 

A progressive muscle dysfunction increases during menopausal transition. This muscle degeneration is caused by the increased levels of inflammatory markers, the decreased proliferation of muscle satellite cells, and changed levels of sex hormones [36]. Particularly, estradiol is essential in skeletal muscle function. Estradiol positively affects skeletal muscle by increasing the proliferation of satellite cell [37]. In addition to the effect, estradiol can limit inflammatory damage on skeletal [37]. Comprehensively, the prevalence of sarcopenia and diabetes increases in menopausal women than in premenopausal women. In addition, it can be inferred that the association was highly affected by confounders during menopause. Menopause leads to the considerable decrease in oestrogen concentrations and it is followed by alterations in energy expenditure, and weight as well as insulin secretion, and insulin sensitivity, which means confounding factors can greatly influence the development of DM with the decrease in oestrogen effect [38]. Moreover, the increased risk of DM during menopause is independent of aging [39,40]. Thereby, the association was insignificant in postmenopausal women after adjusting for confounding factors, even though the association in the same age group of premenopausal women was significant.

Despite the recruitment of large number of participants, there are several limitations to our study. First, we could not differentiate between type 1 and type 2 diabetes because there was no information on serum insulin, C-peptide, and pancreatic autoantibodies in the current KNHANES data. Second, we cannot confirm whether the association between handgrip strength and DM is independent of muscle mass, as there are no muscle data in the KNHANES data. Nevertheless, we used handgrip strength because it is a more useful indicator than muscle mass [12]. Sarcopenia is considered muscle failure, with low muscle strength superior to a lack of muscle mass, which suggests that muscle strength is a primary indicator of sarcopenia [41,42]. Third, the causality between handgrip strength and diabetes cannot be demonstrated because it is a cross-sectional study. Moreover, it cannot explain why the association of sarcopenia with the prevalence of diabetes in postmenopausal women was insignificant. Additional research is needed on the mechanism between muscle dysfunction and diabetes in postmenopausal women. Finally, a suitable index to eliminate the impact of body size (weight, height, and BMI) on handgrip strength has not yet been determined. Albeit we used RGS to minimise the effect of body size, dividing RGS by BMI cannot completely eliminate the effect of body size [43]. Further studies are needed on muscle strength-related indices independent of body size.

## 5. Conclusions

We found that RGS was independently inversely associated with the prevalence of DM in men and premenopausal women. The association of handgrip strength with DM is more significant with the younger group in men, also significant in premenopausal women who are 40–59 years old, and insignificant in postmenopausal women at all age groups. RGS can be a practical tool to predict the prevalence of DM. The appropriate examination of handgrip strength is important for detecting DM.

## Figures and Tables

**Figure 1 diagnostics-12-01874-f001:**
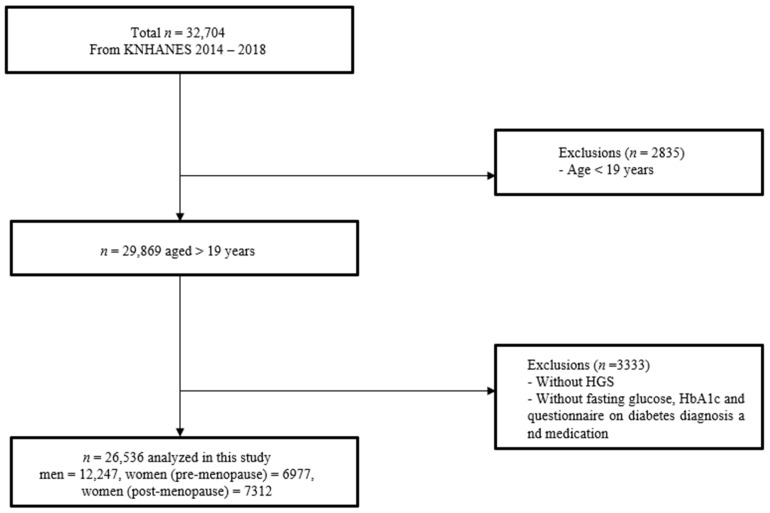
Flow diagram of subjects meeting the inclusion/exclusion criteria.

**Figure 2 diagnostics-12-01874-f002:**
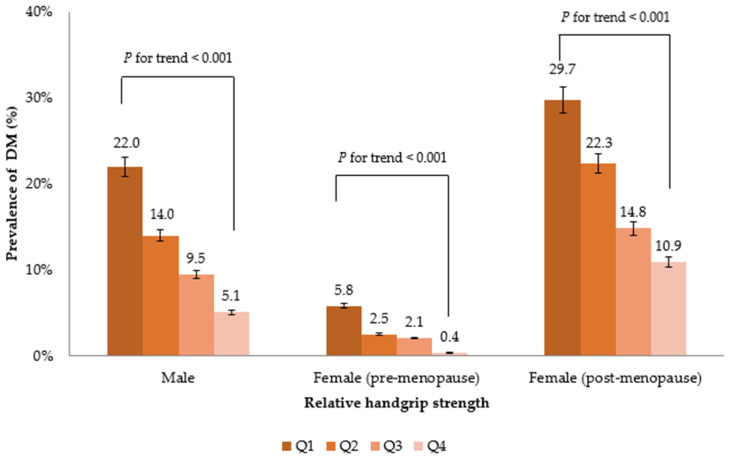
Prevalence of DM according to RGS (GS/BMI) quartiles.

**Table 1 diagnostics-12-01874-t001:** Baseline characteristics of the study population according to the RGS quartile.

	Men	Q_1_	Q_2_	Q_3_	Q_4_
		≤2.84	2.85–3.30	3.31–3.75	>3.75
*n*	12,247	3721	3170	2809	2547
Age (years)	45.7 ± 0.2	52.6 ± 0.4	47.2 ± 0.4	43.7 ± 0.3	39.3 ± 0.3
Waist circumference (cm)	85.8 ± 0.2	91.0 ± 0.2	87.3 ± 0.2	84.6 ± 0.2	80.5 ± 0.2
BMI (kg/m^2^)	24.5 ± 0.1	26.3 ± 0.1	25.0 ± 0.1	24.0 ± 0.1	22.6 ± 0.1
Fasting glucose (mg/dL)	102.1 ± 0.3	108.3 ± 0.6	103.7 ± 0.5	100.7 ± 0.6	95.9 ± 0.4
Total cholesterol (mg/dL)	190.6 ± 0.4	189.1 ± 0.8	193.1 ± 0.8	192.3 ± 0.8	187.9 ± 0.8
Triglyceride (mg/dL)	164.4 ± 1.7	167.9 ± 2.7	181.1 ± 3.3	163.4 ± 3.3	145.5 ± 3.6
AST (IU/L)	24.9 ± 0.3	27.4 ± 0.4	25.6 ± 0.3	24.1 0.2	22.5 ± 0.3
ALT (IU/L)	27.6 ± 0.4	32.6 ± 0.6	29.7 ± 0.5	26.1 ± 0.4	22.4 ± 0.3
Systolic BP (mmHg)	119.8 ± 0.2	122.9 ± 0.3	120.9 ± 0.3	119.2 ± 0.3	116.3 ± 0.3
Diastolic BP (mmHg)	78.1 ± 0.2	77.7 ± 0.2	78.9 ± 0.2	78.8 ± 0.2	77.2 ± 0.2
Smoking status, *n* (%)					
Never smoker	2776 (25.3)	871 (26.8)	705 (25.1)	635 (25.4)	565 (23.9)
Ex-smoker	4882 (36.1)	1720 (41.9)	1341 (38.6)	1019 (33.7)	802 (30.4)
Current smoker	4248 (38.6)	1011 (31.3)	1054 (36.3)	1071 (40.9)	1112 (45.7)
Alcohol uptake, *n* (%)	3816 (33.9)	923 (27.8)	1028 (35.6)	988 (36.4)	880 (35.9)
Regular exercise, *n* (%)	3386 (32.7)	736 (24.3)	869 (31.0)	890 (35.9)	891 (39.8)
Hypertension, *n* (%)	4178 (29.4)	1813 (43.4)	1154 (32.7)	747 (24.5)	464 (16.8)
	Women (premenopause)	Q_1_	Q_2_	Q_3_	Q_4_
		≤1.90	1.91–2.22	2.23–2.53	>2.53
N	6977	1757	1749	1765	1706
Age (years)	35.4 ± 0.1	36.1 ± 0.3	35.6 ± 0.3	35.5 ± 0.3	34.5 ± 0.3
Waist circumference (cm)	75.4 ± 0.3	81.3 ± 0.3	76.4 ± 0.3	73.3 ± 0.2	70.1 ± 0.2
BMI (kg/m^2^)	22.6 ± 0.1	25.3 ± 0.1	23.1 ± 0.1	21.7 ± 0.1	20.4 ± 0.1
Fasting glucose (mg/dL)	92.3 ± 0.2	95.8 ± 0.6	92.7 ± 0.5	90.8 ± 0.3	89.8 ± 0.2
Total cholesterol (mg/dL)	186.8 ± 0.5	193.4 ± 1.1	187.8 ± 0.8	185.7 ± 0.9	180.1 ± 0.8
Triglyceride (mg/dL)	96.6 ± 0.9	115.3 ± 2.2	99.3 ± 1.9	92.1 ± 1.7	80.2 ± 1.3
AST (IU/L)	18.7 ± 0.2	20.3 ± 0.3	18.3 ± 0.2	18.2 ± 0.2	17.8 ± 0.2
ALT (IU/L)	15.4 ± 0.3	19.0 ± 0.5	15.1 ± 0.3	14.3 ± 0.3	13.2 ± 0.2
Systolic BP (mmHg)	107.6 ± 0.2	109.7 ± 0.4	107.8 ± 0.3	106.6 ± 0.3	106.1 ± 0.3
Diastolic BP (mmHg)	71.6 ± 0.3	73.2 ± 0.2	71.7 ± 0.3	70.9 ± 0.3	70.7 ± 0.3
Smoking status, *n* (%)					
Never smoker	5941 (84.9)	1490 (84.1)	1496 (85.5)	1501 (85.5)	1454 (84.5)
Ex-smoker	569 (8.1)	141 (8.2)	132 (7.3)	159 (8.4)	137 (8.4)
Current smoker	458 (7.0)	124 (7.7)	119 (7.2)	101 (6.1)	114 (7.1)
Alcohol uptake, *n* (%)	914 (13.5)	249 (14.5)	239 (13.6)	194 (11.4)	232 (14.5)
Regular exercise, *n* (%)	2023 (29.6)	426 (23.8)	482 (28.7)	559 (32.3)	556 (33.4)
Hypertension, *n* (%)	467 (6.1)	189 (10.2)	128 (6.3)	85 (4.5)	65 (3.4)
	Women (postmenopause)	Q_1_	Q_2_	Q_3_	Q_4_
		≤1.48	1.49–1.81	1.82–2.13	>2.13
N	7312	1938	1897	1787	1690
Age (years)	63.3 ± 0.2	69.4 ± 0.3	64.5 ± 0.3	61.4 ± 0.2	57.6 ± 0.2
Waist circumference (cm)	81.8 ± 0.3	86.3 ± 0.3	84.0 ± 0.3	80.8 ± 0.2	76.2 ± 0.2
BMI (kg/m^2^)	24.1 ± 0.1	25.8 ± 0.1	24.9 ± 0.1	23.8 ± 0.1	22.1 ± 0.1
Fasting glucose (mg/dL)	103.4 ± 0.4	108.0 ± 0.8	105.3 ± 0.7	101.5 ± 0.6	99.3 ± 0.6
Total cholesterol (mg/dL)	199.5 ± 0.6	193.1 ± 1.2	198.4 ± 1.2	203.5 ± 1.1	202.4 ± 1.0
Triglyceride (mg/dL)	132.0 ± 1.3	141.8 ± 2.6	138.6 ± 2.6	129.7 ± 2.5	118.6 ± 2.3
AST (IU/L)	23.4 ± 0.3	23.9 ± 0.3	24.2 ± 0.4	23.2 ± 0.2	22.4 ± 0.2
ALT (IU/L)	20.4 ± 0.4	20.4 ± 0.4	21.6 ± 0.5	20.8 ± 0.3	18.8 ± 0.3
Systolic BP (mmHg)	123.7 ± 0.3	127.8 ± 0.5	125.6 ± 0.5	122.6 ± 0.5	118.8 ± 0.5
Diastolic BP (mmHg)	74.7 ± 0.3	73.4 ± 0.3	75.0 ± 0.3	75.4 ± 0.3	75.0 ± 0.3
Smoking status, *n* (%)					
Never smoker	6749 (92.8)	1770 (92.3)	1773 (94.4)	1646 (92.5)	1560 (92.1)
Ex-smoker	247 (3.4)	75 (4.1)	62 (3.0)	66 (3.6)	44 (2.8)
Current smoker	245 (3.8)	68 (3.6)	41 (2.6)	60 (4.0)	76 (5.1)
Alcohol uptake, *n* (%)	319 (5.1)	46 (2.7)	83 (5.4)	95 (5.9)	95 (6.4)
Regular exercise, *n* (%)	1109 (16.2)	163 (8.4)	238 (13.5)	304 (18.1)	404 (24.8)
Hypertension, *n* (%)	3567 (45.6)	1246 (63.5)	1038 (50.2)	775 (40.4)	508 (28.2)

RGS, relative handgrip strength; BMI, body mass index; AST, aspartate aminotransferase; ALT, alanine aminotransferase.

**Table 2 diagnostics-12-01874-t002:** Association between RGS (per 0.1 kg) and the prevalence of DM in Koreans using multivariate logistic regression.

Men			Women (Premenopause)			Women (Postmenopause)		
	OR	*p*-Value		OR	*p*-Value		OR	*p*-Value
Unadjusted	0.41 (0.37–0.45)	<0.001	Unadjusted	0.21 (0.15–0.28)	<0.001	Unadjusted	0.36 (0.31–0.42)	<0.001
Model 1	0.59 (0.53–0.67)	<0.001	Model 1	0.23 (0.17–0.32)	<0.001	Model 1	0.50 (0.42–0.59)	<0.001
Model 2	0.79 (0.68–0.91)	0.001	Model 2	0.65 (0.46–0.93)	0.018	Model 2	0.80 (0.66–0.98)	0.027
Model 3	0.84 (0.74–0.95)	0.007	Model 3	0.68 (0.47–0.97)	0.035	Model 3	0.83 (0.68–1.01)	0.058

RGS, relative handgrip strength; SE, standard error; Model 1, adjusted for age; Model 2, adjusted for age, waist circumference, regular exercise, smoking status, and alcohol uptake; Model 3, adjusted for age, waist circumference, regular exercise, smoking status, alcohol uptake, total cholesterol, TG, AST, ALT, and systolic blood pressure.

**Table 3 diagnostics-12-01874-t003:** Odds ratio and 95% confidence intervals for the prevalence of DM according to RGS quartile.

	Men				Women (Pre-Menopause)				Women (Post-Menopause)			
	Q_1_	Q_2_	Q_3_	Q_4_	Q_1_	Q_2_	Q_3_	Q_4_	Q_1_	Q_2_	Q_3_	Q_4_
	≤2.84	2.85–3.30	3.31–3.75	>3.75	≤1.90	1.91–2.22	2.22–2.53	>2.53	≤1.48	1.48–1.81	1.81–2.13	>2.13
Unadjusted	1.00	0.56 (0.48–0.66)	0.39 (0.32–0.46)	0.16 (0.13–0.21)	1.00	0.42 (0.29–0.62)	0.35 (0.22–0.54)	0.07 (0.03–0.15)	1.00	0.68 (0.57–0.81)	0.41 (0.33–0.50)	0.29 (0.23–0.36)
Model 1	1.00	0.72 (0.61–0.85)	0.60 (0.50–0.72)	0.32 (0.25–0.41)	1.00	0.44 (0.30–0.66)	0.38 (0.24–0.59)	0.08 (0.04–0.18)	1.00	0.81 (0.67–0.98)	0.54 (0.44–0.67)	0.44 (0.34–0.56)
Model 2	1.00	0.88 (0.75–1.04)	0.85 (0.70–1.04)	0.54 (0.42–0.71)	1.00	0.78 (0.50–1.22)	1.03 (0.62–1.71)	0.30 (0.14–0.66)	1.00	0.92 (0.75–1.12)	0.75 (0.60–0.94)	0.81 (0.62–1.05)
Model 3	1.00	0.87 (0.73–1.04)	0.90 (0.73–1.10)	0.57 (0.43–0.75)	1.00	0.82 (0.52–1.30)	1.10 (0.65–1.86)	0.33 (0.14–0.75)	1.00	0.91 (0.73–1.12)	0.77 (0.61–0.97)	0.82 (0.63–1.07)

Model 1: adjusted for age; Model 2: adjusted for age, waist circumference, regular exercise, smoking status, and alcohol uptake; Model 3: adjusted for age, waist circumference, regular exercise, smoking status, alcohol uptake, total cholesterol, TG, AST, ALT, and systolic blood pressure.

## Data Availability

The data underlying this article will be shared upon reasonable request from the corresponding author.

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
