# Peer review of "Association of Handgrip Strength with Diabetes Mellitus in Korean Adults According to Sex"

_diagnostics, 2022, doi:10.3390/diagnostics12081874_

Round 1
Reviewer 1 Report
• As the prevalence of DM has increased significantly around the world in recent years, it is indeed important to identify early in order to formulate an intervention plan to prevent or exacerbate the disease and, yes, reduce its harmful effects on chronic diseases.
• It is important in this type of study to neutralize differences in muscle mass that depend on gender or body structure and this may have been the reason for conflicting findings in previous studies. Disabling BMI is not enough because it does not express body size. Was correct to neutralize according to an index that expresses total muscle mass and is not neutralized in relation to height. Defining a body type by the respondents could also have been relevant. It may even be necessary to perform a separate analysis by gender and verify the results between the genders as there are enough observations for research intensity for a separate analysis by gender.
• It is not clear, why methodologically, should have been divided into quarters while losing information and not using the average of the three measurements of hand grip strength according to the actual continuous result? Why is it not more correct to take the maximum strength from the three measurements?
• Hand grip strength is associated with residual life expectancy and it is clear that as you get older it is the dominant factor. Since the aim of the study is early prediction of DM, it was practical to focus on younger ages. In older age when the incidence of Sarcopenia increases, the effectiveness of early detection and early treatment decreases.
Reviewer 2 Report
Association between hand-grip strength and diabetes mellitus have been examined in large number of participants in this manuscript. Work seems interesting however more rigorous data analysis and its reporting is needed particularly identification of effect modifiers in addition to confounders and stratification thereon.
There are few other suggestions
1)Title is not clear, two concepts are mixed, reframing is needed.
2)Abstract- Research question is not clearly stated
Authors have used word ‘prevalence’ for indicating presence of diabetes, however readers may confuse it with the percentage of participants having diabetes.
3) Methods- What was dependent variable, if it is presence or absence of diabetes, binary logistic regression is appropriate term instead of multinominal logistic regression.
4) Results- Age seems to be confounder as well as effect modifier and thus age stratified analysis should be done instead of only age-adjustment
5) Model diagnostics i.e. whether assumptions were met or not, Goodness-of-Fit, pseuso-R2 etc to be presented. It may be presented as supplementary table.
Round 2
Reviewer 2 Report
Please thoroughly respond/revise as per previous comments
Word 'risk' is an inappropriate replacement for word 'prevalent'
Model diagnostics of logistic regression doesn't only mean Pseudo-R2 other diagnostics needs to be presented. Also looking at R2 , effect size seems minimal.
Supplementary Table shows results stratified by age to examined effect modification. Presence or absence of effect modification needs to be explained in the main text along with its implications.
